# Evidence of Metabolic Dysfunction in Amyotrophic Lateral Sclerosis (ALS) Patients and Animal Models

**DOI:** 10.3390/biom13050863

**Published:** 2023-05-19

**Authors:** Katarina Maksimovic, Mohieldin Youssef, Justin You, Hoon-Ki Sung, Jeehye Park

**Affiliations:** 1Genetics and Genome Biology Program, The Hospital for Sick Children, Toronto, ON M5G 0A4, Canada; katarina.maksimovic@sickkids.ca (K.M.); mohieldin.youssef@sickkids.ca (M.Y.); justin.you@mail.utoronto.ca (J.Y.); 2Translational Medicine Program, The Hospital for Sick Children, Toronto, ON M5G 0A4, Canada; hoon-ki.sung@sickkids.ca; 3Department of Laboratory Medicine & Pathobiology, University of Toronto, Toronto, ON M5S 1A8, Canada; 4Department of Molecular Genetics, University of Toronto, Toronto, ON M5S 1A8, Canada

**Keywords:** amyotrophic lateral sclerosis, metabolic dysfunction, hypermetabolism

## Abstract

Amyotrophic lateral sclerosis (ALS) is a neurodegenerative disease that affects motor neurons, leading to muscle weakness, paralysis, and eventual death. Research from the past few decades has appreciated that ALS is not only a disease of the motor neurons but also a disease that involves systemic metabolic dysfunction. This review will examine the foundational research of understanding metabolic dysfunction in ALS and provide an overview of past and current studies in ALS patients and animal models, spanning from full systems to various metabolic organs. While ALS-affected muscle tissue exhibits elevated energy demand and a fuel preference switch from glycolysis to fatty acid oxidation, adipose tissue in ALS undergoes increased lipolysis. Dysfunctions in the liver and pancreas contribute to impaired glucose homeostasis and insulin secretion. The central nervous system (CNS) displays abnormal glucose regulation, mitochondrial dysfunction, and increased oxidative stress. Importantly, the hypothalamus, a brain region that controls whole-body metabolism, undergoes atrophy associated with pathological aggregates of TDP-43. This review will also cover past and present treatment options that target metabolic dysfunction in ALS and provide insights into the future of metabolism research in ALS.

## 1. Introduction

Amyotrophic lateral sclerosis (ALS) was first defined by Jean-Martin Charcot in 1874 when lesions in the spinal cord were identified in autopsies of patients with progressive paralysis [1]. Since then, many research groups have focused primarily on investigating the causes and mechanisms underlying motor neuron death. The onset of ALS ranges from 40 to 60 years old, with death typically occurring 2 to 5 years after onset, often due to respiratory failure [2]. Approximately 10% of cases are familial, while the remaining 90% of cases are sporadic [2]. Mutations in over 30 genes involved in various cellular processes have been linked to ALS, including proteostasis, RNA metabolism, and cytoskeletal dynamics [3]. Some of the more commonly studied ALS-linked genes are superoxide dismutase (*SOD1*), transactive response DNA-binding protein 43 (*TARDBP*), fused in sarcoma (*FUS*), and chromosome 9 open reading frame 72 (*C9ORF72*) [3]. Decades of research on the genetics and neuropathology of ALS and the use of animal models have significantly advanced our understanding of the disease [4]. Accumulating studies (particularly those with ALS-linked mutations in *SOD1* and *TDP-43*) have shown that ALS is not simply a disease of motor neurons, but rather a complex, multisystemic, and multifactorial disease that involves many other neuronal types, glial cells, and peripheral tissues [4].

Early clinical indicators of ALS have enhanced our understanding of the disease pathogenesis and provided clues on prognostic factors that may affect the course and outcome of ALS patients. Early clinical signs include weight loss, metabolic disturbances, and hypermetabolism [5,6,7]. Hypermetabolism refers to an excessive increase in measured resting energy expenditure (REE) when compared to the predicted REE [7]. Hypermetabolism is associated with increased catabolism of carbohydrates, proteins, and lipids, ultimately leading to weight loss and muscle wasting [7]. Importantly, accumulating evidence has revealed that ALS patients show muscle atrophy and weight loss before the loss of motor neurons and the onset of neurological symptoms [8,9], supporting the hypothesis that metabolic dysfunction or hypermetabolism may play a pivotal role in the early stages of ALS development. This review will examine recent advances in research on the multisystemic metabolic dysfunction that contributes to ALS development and progression.

## 2. Metabolic Dysfunction in ALS Patients and Animal Models

Reduced body weight or body mass index (BMI) is an independent prognostic factor for ALS and often precedes the onset of the disease by decades [9]. Lower BMIs in ALS patients are correlated with worsened motor symptoms and shorter survival [10,11,12]. Additionally, a population-based study demonstrated that subjects with high BMI are associated with a reduced risk of developing ALS several decades later [13], further suggesting a role of metabolism in the pathogenesis of ALS. The potential causes of weight loss in ALS are multifactorial and include hypermetabolism as well as malnutrition resulting from a loss of appetite, independent of swallowing difficulties (dysphagia) [14,15,16]. Dysphagia is an early-onset symptom in patients with bulbar-onset ALS, while the majority of patients with spinal-onset ALS develop dysphagia as disease progresses [17], which can exacerbate reduced caloric intake. ALS patients, even without dysphagia, generally have increased REE or hypermetabolism [5,18]. REE accounts for approximately 60–75% of total energy expenditure and is dependent on body composition and sex [19]. Hypermetabolism is observed in about half of all ALS patients, irrespective of sex, and is correlated with reduced body weight and BMI, shorter survival, and greater functional decline [5,6,18,20,21]. However, the mechanisms underlying hypermetabolism and its origins in ALS remain elusive.

Accumulating studies with animal models provide clues as to the origin of hypermetabolism in ALS. Most ALS mouse models that exhibit clinically relevant motor phenotypes also show impaired weight gain phenotype (Table 1), indicating that metabolic dysfunction is associated with ALS pathogenesis. Hypermetabolism has also been observed in these animal models at the presymptomatic stage or at disease onset, suggesting that hypermetabolism underlies impaired weight gain and disease development and progression. These mouse models have been further characterized in various metabolic aspects to dissect the underlying cause of hypermetabolism. Some ALS mouse models have also exhibited reduced fat-free mass (or reduced muscle mass), reduced fat mass, dyslipidemia, reduced glucose utilization, liver and pancreas dysfunction, hypothalamic neurodegeneration, and increased food intake, usually after fasting, showing additional metabolic features of ALS (Table 1). In this review, we will focus on discussing dysregulated metabolic aspects, particularly in the CNS and peripheral metabolic organs in both ALS patients and animal models, as there are many other reviews that discuss metabolic dysregulation in motor neurons.

## 3. Reduced Fat-Free Mass (or Reduced Muscle Mass) and Reduced Glucose Utilization

Muscle wasting is a key feature of ALS, and is reflected in the reduction of fat-free mass [51,52]. Similar to ALS patients, many ALS mouse models display muscle atrophy or reduced muscle mass (Table 1) [31,40,43,47].

Skeletal muscle acquires ATP through three different pathways, depending on the intensity of exertion: the glycolytic system, phosphagen system (also known as the creatine metabolism pathway), and mitochondrial respiration (which encompasses fatty acid oxidation) [85]. During bouts of high-intensity exercise, energy is supplied via the creatine metabolism pathway. Creatine is transported to the muscles via the blood, where it is converted into phosphocreatine by the enzyme creatine kinase (CK), acting as a reserve of phosphates for ATP production [86]. Creatinine is a byproduct of this pathway that is expelled through the urine, and its levels are correlated with fat-free mass [87]. Serum CK and creatinine levels have been examined in ALS patients for decades, and overall, there are inconsistencies regarding whether CK and creatinine levels are increased or decreased compared to healthy controls. There is some debate as to whether increased levels are positively associated with survival [88,89,90,91]. A recent study showed that serum creatinine, rather than CK, is a better predictive factor of survival in ALS patients [91]. Elevating levels of creatine to counter energy dysregulation has been a focus of clinical trials in the past, with dietary creatine supplements failing to reveal therapeutic benefits in ALS patients [92,93].

Muscles are not equal in their composition or energy demand, and are not equally affected in ALS [94]. Fast-twitch muscle fibres, which generally use glycolysis as their main energy source, are preferentially affected in ALS, whereas the ALS-resistant slow-twitch muscle fibres have a high oxidative capacity [95,96,97]. This energy preference in the context of a hypermetabolic state may be underlying the selective vulnerability of certain muscle fibre types. In a *SOD1^G93A^* mouse model of ALS, muscles mainly comprising fast-twitch fibres are affected first, exhibiting a reduction in contractile force and NMJ innervation prior to slow-twitch [95,98]. A recent study reported that glycolytic muscles in *SOD1^G93A^* mice and myotubes from ALS patients show increased dependence on fatty acid oxidation [31]. In addition, a study using a *SOD1^G86R^* mouse model performed gene expression profiling and demonstrated a fuel preference switch from glycolysis to fatty acid oxidation in the fast-twitch tibialis anterior muscle [69]. This study also showed that promoting glycolytic capacity by administering dichloroacetate improved muscle strength [69]. These findings are consistent with another study that used Ranolazine, an inhibitor of fatty acid oxidation used to promote glycolysis, which improved muscle strength and ATP content in the muscle [49]. In addition, this study described mitochondrial alterations and reduced ATP levels specific to glycolytic muscles at the early symptomatic stage [49]. Since mitochondrial dysfunction plays a role in the energy imbalance (see the section on “Metabolic Dysregulation in the Central Nervous System” as well), overexpressing genes that improve mitochondrial function (e.g., PGC-1α in the muscle of *SOD1* mouse models) was able to rescue muscle function and locomotor phenotypes [99].

## 4. Reduced Fat Mass and Dyslipidemia

A number of studies have reported a reduction in fat mass in ALS patients and animal models (Table 1) [20,31,32,51,52]. Fat can differ based on its location throughout the body, with visceral fat deposited around the internal organs, and subcutaneous fat beneath the skin. In the context of ALS, one study examined body fat distribution using MRI and found that ALS patients have increased visceral fat and a non-significant trend of decreased subcutaneous fat [53]. The amount of subcutaneous fat was correlated with survival in male ALS patients but not females [53]. Visceral fat may not be correlated with survival in this study as a high amount of visceral fat reflects insulin resistance and is associated with systemic inflammation, whereas increased subcutaneous fat may be protective as it is reflective of increased energy storage [100,101].

Adipose tissue is the central storage site of energy in the form of triacylglycerols [102]. Adipose tissue is also highly sensitive to insulin, which stimulates glucose uptake via GLUT4, leading to lipogenesis. Increased rates of lipolysis (hydrolysis of triacylglycerols) will occur during states of energy deprivation to create glycerol and free fatty acids, as glycerol can be used for gluconeogenesis by the liver, and free fatty acids can be used for oxidative phosphorylation by the muscle [102]. The two major types of mammalian fat tissues are white adipose tissue (WAT) and brown adipose tissue (BAT). WAT is involved in energy storage and release and hormone secretion and comprises the majority of all adipose tissue [103]. BAT has a high number of mitochondria that express uncoupling protein 1 (UCP1), which enables the maintenance of body temperature via thermogenesis [104]. An intermediate between WAT and BAT is beige adipocytes, which reside in WAT but exhibit characteristics of brown adipocytes. Formation of beige adipocytes in WAT (i.e., browning of WAT) can be induced by conditions such as cold environment, exercise, and hypermetabolism [102,103,105]. Studies using *SOD1^G93A^* mice demonstrated elevated lipolysis [31,106] and browning of WAT [106], suggestive of a hypermetabolic state.

Dyslipidemia, referring to an imbalance in lipids in the blood, has been observed in ALS patients with some contention. There have been a number of studies that have examined a variety of circulating lipids in ALS patient blood and cerebrospinal fluid (CSF) with sometimes inconsistent findings on trends and correlation with survival, but overall there are trends of increased levels (hyperlipidemia) of fatty acids, sterol lipids, sphingolipids, triglycerides, and glycerophospholipids (see [61,62] for a comprehensive review). One lipidomics study examined the CSF of 40 ALS patients and determined a specific lipidomic signature which was also confirmed in *SOD1^G93A^* mice [58]. This study was also matched certain lipids to better prognosis and survival, such as higher levels of sphingomyelins and lower levels of triglycerides [58]. Another study that measured plasma lipids at baseline and followed up 2 years later found that ALS patients exhibit progressive alterations in lipid composition, particularly in sphingolipid and glycerophospholipid classes [59]. A lipidomics study in *SOD1^G93A^* rats at symptomatic stages found lipid profile alterations specific to disease in the spinal cord, such as decreased levels of cardiolipin and a dramatic increase in several cholesteryl esters linked to polyunsaturated fatty acids [107]. Elevated lipid levels in the CSF and blood [61,62] may be indicative of an energy source preference switch. Hyperlipidemia may be beneficial, as increased levels of low-density lipoprotein cholesterol (LDL-C), high-density lipoprotein cholesterol (HDL-C), apolipoproteins AI and B, and serum triglyceride levels were associated with increased survival in ALS patients [55,56,108]. However, these findings contrast with a previous study in which high levels of LDL and HDL were not associated with survival when corrected for BMI and age, likely owing to the heterogeneity in ALS disease progression [22]. The inconsistencies in some studies may be explained by their methodology. The timing of the study is important to consider, as there can be major differences if samples are taken at diagnosis or later in disease. Furthermore, it is unclear whether lipid profiles from serum and CSF should be directly compared. More studies are needed to elucidate and confirm the correlation of lipid profiles in serum and CSF with consideration to disease duration and BMI.

## 5. Liver and Pancreas Dysfunction

The liver is a central player in the metabolism of carbohydrates, fat, and proteins, as well as the detoxification of the blood. Despite the vast, important roles of the liver, it has been underappreciated in the field of ALS research. An early study reported a high incidence of liver dysfunction in ALS patients and found structural changes in hepatocytes such as abnormal mitochondria and endoplasmic reticula [73]. Another study using a *SOD1^G93A^* mouse model reported inflammation and atrophy in the liver at disease end-stage, as well as a reduction in IGF-1 levels, a liver-derived growth factor that is important for neuronal survival [75]. A more recent study found that fatty liver disease is highly prevalent in ALS, affecting 76% of patients, which was significantly higher compared to asymptomatic dyslipidemic controls and non-ALS neurodegenerative disease patients [74]. These studies exemplify that dysregulated metabolism in the liver may contribute to dyslipidemia in ALS patients. Given the role of the liver in energy homeostasis, a few papers studied the role of the cellular energy sensor, AMP-activated protein kinase (AMPK), in ALS. AMPK is a central regulator of energy sensing and metabolic coordination and is activated by an imbalance in ATP and metabolic stresses [109]. AMPK initiates an activation cascade to replenish cellular ATP by increasing fatty acid oxidation and glucose uptake, while reducing cellular anabolic pathways, such as protein and glycogen synthesis [110]. Increased AMPK activation has been reported in a *SOD1^G93A^* mouse line [111]; however, another study found no difference in AMPK activation in the liver when compared to the non-transgenic control [112], which may be due to differences in mouse strain background and sex of mice used for experiments.

The pancreas is an organ that produces important enzymes required for digestion and hormones that regulate blood glucose, such as insulin, glucagon, and somatostatin. Like the liver, the pancreas has not gathered much attention in ALS research in the past but has recently gained appreciation for its contribution to metabolic dysregulation in ALS. ALS patients show pancreatic dysfunction and impaired glucose tolerance [113,114,115], which may arise due to abnormalities in the release or the response to insulin and glucagon. A recent study reported reduced early-phase insulin secretion in ALS patients, and autopsies of the pancreas β-cells from sporadic ALS patients demonstrated a loss of nuclear TDP-43, a key neuropathological hallmark in ALS, although there was no reduction in β-cell mass [76]. In contrast to the finding in ALS patients, the *SOD1^G93A^* mouse model also showed dysfunction in the pancreas, with the loss of insulin-positive β-cells rather than a functional impairment [77]. How dysfunction of the liver and pancreas contributes to metabolic dysregulation or hypermetabolism in ALS patients needs to be further investigated.

## 6. Metabolic Dysregulation in the Central Nervous System (CNS)

The brain requires a relatively higher amount of energy compared to other organs. Despite accounting for only 2% of body weight, the brain utilizes 20% of the body’s energy output [116,117]. It is estimated that 75–80% of the energy used by the brain is mainly consumed by neurons [118]. Neuronal energy is crucial for supporting the integrity of long axons, synaptic activity, and facilitating action potentials [119,120,121]. The primary energy source for the CNS is glucose. Therefore, abnormalities in glucose metabolism can have a significant impact on CNS health and viability [122]. Fluorodeoxyglucose-positron emission tomography (FDG-PET) is a valuable clinical tool to measure glucose metabolism in the brain. FDG-PET imaging studies have shown that ALS patients exhibit brain-region specific metabolic defects, with hypometabolism in the frontal, motor, and occipital cortex, and hypermetabolism in the midbrain, temporal pole, and hippocampus [67] (Table 1). In animal models of ALS, glucose utilization levels were lower in the CNS. In the brain and spinal cord of *SOD1^G93A^* mice, glucose utilization levels were decreased even prior to symptom onset [68,70]. Additionally, lowered glucose metabolism was observed in the motor and somatosensory cortices of *TDP-43^A315T^* mice [72]. A decrease in glucose transport can be one of the factors underlying decreased utilization, as GLUT1, a major glucose transporter in the CNS, was found to be decreased in the endothelial cells of the spinal cord in *SOD1^G93A^* mice [71]. A study on an ALS model of *Drosophila* overexpressing mutant TDP-43 in motor neurons exhibited alterations in glucose metabolism. Increasing neuronal glucose availability rescued locomotor deficits, which was achieved by feeding a high-sugar diet or by overexpressing the human glucose transporter GLUT3 in motor neurons [123]. These studies indicate that glucose transport and glycolysis pathway are affected in ALS.

The abnormal glucose metabolism and insufficient energy production would, indeed, affect lipid metabolism. Ceramides and cholesterol esters were increased in ALS patients and a *SOD1^G93A^* mouse model [124]. Additionally, a *Drosophila* model of ALS overexpressing *TDP-43^G298S^* in motor neurons demonstrated altered mitochondrial carnitine shuttling and decreased levels of a ketone precursor produced from lipid oxidation [125]. Overexpression of *TDP-43^A315T^* in the CNS of mice inhibited cholesterol biosynthesis [126]. A recent study demonstrated that elevated levels of arachidonic acid were observed in ALS patient iPSC-derived cultures, and *Drosophila* and mouse models of ALS, and reducing the levels of arachidonic acid was sufficient to reverse the phenotypes in these models [127]. These studies demonstrate that abnormal neuronal lipid metabolism is implicated in ALS pathogenesis.

The mitochondria are the primary suppliers of cellular energy and have been found to exhibit defects in ALS. An abnormal phenotype of vacuolated mitochondria has been observed in mutant *SOD1* mice [30]. Mitochondrial function assessed by mitochondrial respiratory chain activities was defective in the spinal cord of the mutant *SOD1* mice [128,129]. In addition, impairment in mitochondrial function was identified in another ALS model that harbors the *FUS^P525L^* mutation [130]. Along with this notion, a decrease in mitochondrial size associated with altered mitochondrial RNA translation has been observed in mouse ESC-derived neurons with the *FUS^R495X^* mutation [131]. Mutations in TDP-43 alter mitochondrial morphology and induce a release of mitochondrial DNA into the cytoplasm, leading to a subsequent inflammatory response [132,133]. Altered mitochondrial functions have also been observed as a result of *C9ORF72* hexanucleotide repeat expansions [134,135,136]. These studies suggest that mutations associated with ALS induce abnormal mitochondrial morphology and function. Furthermore, a number of studies have shown that alterations in mitochondrial metabolism and function lead to elevated oxidative stress [137,138,139,140,141]. These studies indicate that metabolic dysfunction resulting from mitochondrial defects and loss of cellular protective mechanisms against oxidative stress can predispose to the development and progression of ALS.

## 7. Dysfunction of the Hypothalamus, the Hub Regulating Whole-Body Metabolism

To maintain energy balance, the calories from food intake must be equal to the three components of energy expenditure: physical activity expenditure, REE, and the thermic effect of food [142]. The brain is mainly responsible for regulating energy balance by controlling appetite and satiety through complex endocrine, metabolic, and neural circuits. Although a broad diversity and distribution of neuronal networks are involved in regulating energy intake and expenditure, the hypothalamus has been recognized as the primary brain region to coordinate these processes. Interestingly, there has been increasing evidence of hypothalamic pathology in ALS patients and animal models (Table 1) [78,79,80,82]. A recent paper reported that hypothalamic connectivity is altered in both ALS patients and a mouse model of ALS [82]. Autopsy studies have also shown that the hypothalamus in ALS patients shows an accumulation of pathological TDP-43 protein inclusions [143,144]. Furthermore, the presence of TDP-43 pathology in the lateral hypothalamus was associated with a lower BMI [143]. These TDP-43 inclusions corresponded with a loss of oxytocin-expressing neurons and a loss of orexin-producing neurons [144], both of which are involved in regulating energy balance.

There have been several reports of hypothalamic atrophy in ALS patients (Table 1) [78,79,80]. Among the largest cohorts to date measuring hypothalamic volume in ALS was an MRI study comprising 251 sporadic ALS cases, 19 symptomatic and 32 presymptomatic mutation carriers, and 112 healthy controls [78]. Severe atrophy of the hypothalamus was observed in sporadic ALS, symptomatic ALS-mutation carriers, and presymptomatic ALS-mutation carriers compared to healthy controls. In addition, atrophy of the hypothalamus was correlated with decreased BMI, and decreased anterior hypothalamic volume was correlated with earlier onset of ALS [78]. Three other independent studies within the past year have also measured hypothalamic volume in ALS patients. In a large Chinese cohort, there was a reduction in hypothalamic volume in ALS patients compared to healthy controls, in which hypothalamic volume was correlated with BMI [79]. In a separate study, lower hypothalamic volume was associated with lower BMI and presented an increased risk of earlier death [80]. However, there were no significant differences in hypothalamic volume in a much smaller cohort of 16 ALS patients and 16 healthy controls, likely owing to the smaller sample size [81]. Together, these results suggest that hypothalamic atrophy occurs in a subpopulation of ALS patients and may correlate with lower BMI, earlier onset, and worsened prognosis. However, more studies with a greater sample size are required to validate these results.

## 8. Treatments Targeting Metabolic Dysfunction

Decades of research have been dedicated to understanding the pathological mechanisms that contribute to ALS, with the hopes of finding a suitable treatment for this deadly disease. So far, three Food and Drug Administration (FDA)-approved drugs are available for ALS patients: Riluzole, which targets neuronal excitotoxicity; Edaravone, which targets oxidative stress; and more recently, Relyvrio, which acts on reducing dysfunction in mitochondria and endoplasmic reticula, providing a modest increase in survival and delay functional decline [145,146,147,148].

Dietary interventions have emerged as a promising treatment avenue to counter metabolic dysfunction and promote energy availability in ALS patients, as dietary changes are easy to implement and are well tolerated with fewer side effects, compared to drug-based treatments. Studies on transgenic mutant mice fed high-fat diets showed that dietary intervention can improve motor function and increase body weight, motor neuron survival, overall survival, and inhibit AMPK activation, which is indicative of improved energy availability [32,149,150,151,152]. However, studies on dietary intervention in ALS patients have provided mixed results; nevertheless, increasing energy availability through a high-calorie, high-fat, or high-carbohydrate diet seems to be beneficial overall [153,154,155,156]. A small clinical study showed that patients fed either a high-carbohydrate/high-calorie or high-fat/high-calorie diet increased survival. However, only the high-carbohydrate/high-calorie diet was significant when compared to the control diet [153]. Another small clinical study found that high-calorie supplements (either high-fat or high-carbohydrate) were able to stabilize body weight when compared to the recorded weight decline prior to the supplementation. However, there was no difference in the rate of functional decline [154]. In a more recent and larger clinical trial, using a high-calorie fatty diet did not demonstrate a significant difference in survival compared to the control diet, but it did show a significant improvement in survival for the fast-progressing subgroup of patients [155]. In addition to supplementing diets with fat or carbohydrates, a study found that milk whey protein supplements were also able to improve weight maintenance and BMI in ALS patients [156]. Interestingly, a study that assessed self-reported diets prior to the development of ALS in a Japanese cohort showed that a high-carbohydrate, low-fat diet was associated with an increased risk of ALS [157]. Overall, these clinical studies demonstrate promise with high-calorie interventions (usually high-fat) as potential ALS treatments. However, the findings do not fully align with studies using ALS mouse models, as patients do not receive treatment until after diagnosis, or well after symptoms have begun, whereas treatments usually began prior to or at the onset of symptoms in mouse studies [32,149,150,151,152]. Therefore, the timing of interventions may be crucial for the efficacy of a treatment and may contribute to the lack of translatability seen between animal and patient studies. Nonetheless, the future seems promising for this avenue of treatment.

## 9. Conclusions and Future Perspectives

This review provides a general overview of the systemic metabolic dysfunction in ALS. Metabolism is an encompassing term that includes chemical reactions and pathways needed to produce energy for life, to the cells, organs, and systems that consume this energy, and the components involved in maintaining energy homeostasis. Energy is determined by the levels of ATP, and any imbalance in one component of ATP-producing pathways can affect a myriad of physiological processes and systems in the human body. In ALS, this imbalance is evident in various tissues and regions of the body (Figure 1), as determined by studies from ALS patients and animal models (Table 1). Patients present metabolic dysfunction with a reduction in BMI, fat mass, and fat-free mass, elevated REE, and altered levels of circulating metabolites, including hyperlipidemia. Vulnerable muscle groups exhibit elevated energy demand and an energy preference switch from glycolysis to fatty acid oxidation. Significant changes occur in adipose tissue with elevated lipolysis. The liver and pancreas exhibit dysfunction that contributes to disturbed glucose homeostasis and insulin resistance. Neurons in the brain and spinal cord show altered glucose transport and metabolism, dysfunctional mitochondria, and oxidative stress, reflective of energy imbalance. The hypothalamus exhibits atrophy with the presence of pathological TDP-43 protein inclusions (Figure 1). Treatments targeting metabolism, including diet intervention, have been tested on ALS patients and animal models, showing promising outcomes.

Considering the interplay of all these components in metabolism, it is difficult to discern whether metabolic dysfunction is the cause or consequence of motor neuron degeneration in ALS. A theory that has been proposed in the field of ALS is that the selective vulnerability of motor neurons may be due to their energy demands not being met. For instance, motor neurons are extremely large cells that must transport signals spanning their long axons, resulting in a high cellular energy demand. However, motor neurons are not equally susceptible to degeneration, and this susceptibility likely depends on intrinsic properties that may affect their energy demand and gene expression profiles, such as target innervation, cell body size and location, and neuronal excitability [158]. α-motor neurons are a subtype of motor neurons that innervate extrafusal muscle fibres and tend to degenerate in ALS, whereas the smaller γ-motor neurons innervate intrafusal fibres of the muscle spindle and tend to be spared [158,159]. This selective vulnerability of motor neuron subgroups may be at least partially attributed to energy demand. A recent study demonstrated that elevated levels of arachidonic acid directly contributed to the degeneration of motor neurons but not the ALS-resistant ocular motor neurons [127], being one of the first studies to demonstrate the concise involvement of metabolic dysfunction underlying selective vulnerability and contributing to neurodegeneration. This theme of selective vulnerability is also evident in muscle tissue, as fast-twitch muscle fibres have an increased energy demand which likely contributes to their atrophy sooner than slow-twitch muscle fibres with a lower energy demand [95,98]. Further investigation is needed to parse out the key metabolic components of pathological mechanisms contributing to neurodegeneration in ALS, which will inform on the role of metabolic dysfunction in neurodegeneration. It is likely that metabolic dysfunction contributes to neurodegeneration along with other pathological mechanisms such as protein aggregation, neuronal excitotoxicity, and neuroinflammation [4]. Based on treatments targeting metabolic dysfunction only providing limited improvement in survival, it is plausible that multiple pathological mechanisms need to be targeted alongside metabolic dysfunction to extend survival even more.

The timing of study is particularly important when considering metabolism, as it can vary greatly from diagnosis/symptom onset to disease end-stage. The majority of studies described in this review examined patients with a disease duration of approximately 2 years, or mice at symptomatic stage (unless specified otherwise). More longitudinal studies of investigating metabolic dysfunction in ALS patients are needed, starting from diagnosis and following up as disease progresses. In parallel, metabolic dysfunction should be examined in mouse models of ALS at presymptomatic and symptomatic stages, as these studies can provide a practical clue as to whether certain features contribute to ALS onset and progression or occur as a result of neurodegeneration.

The importance of metabolism cannot be underestimated in the field of ALS. The future of ALS research means focusing on metabolism and taking metabolic dysfunction into account when investigating ALS animal models and when patients present with ALS-like symptoms in clinic. Only by shedding light on the involvement of metabolism in the development and progression of ALS can key questions be answered on the prominence of metabolic dysfunction and its contribution to initiating and/or exacerbating neurodegeneration.

## Figures and Tables

**Figure 1 biomolecules-13-00863-f001:**
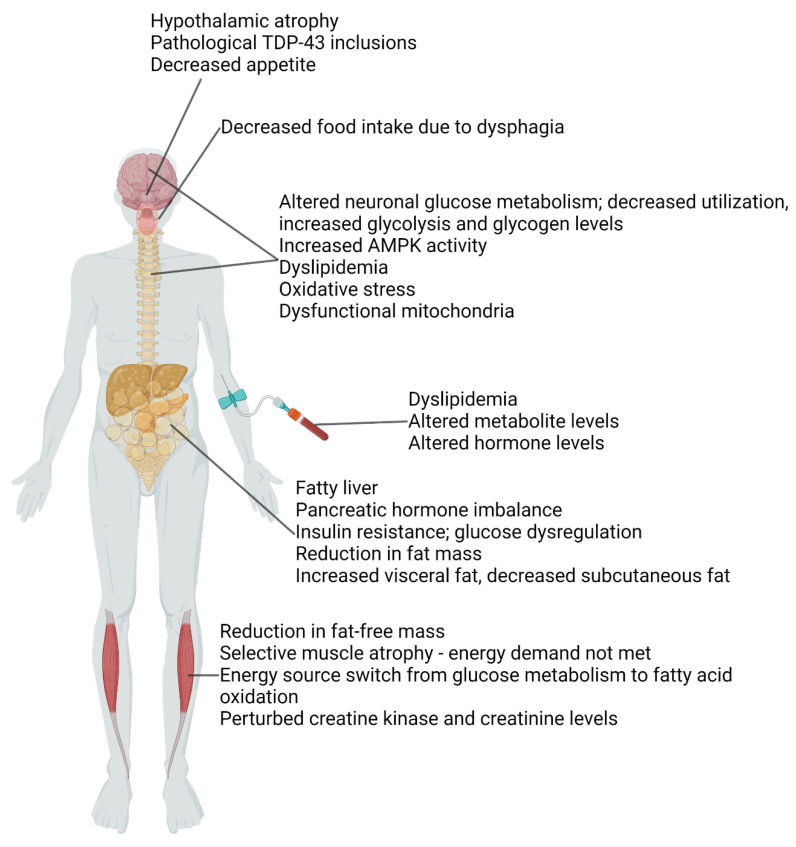
Summary schematic of key findings in ALS patients. Created with BioRender.com (accessed on 24 November 2022).

**Table 1 biomolecules-13-00863-t001:** **Summary of metabolic dysfunction in ALS patients and mouse models.** Tg = transgenic; KI = knock-in. Jackson Laboratory (JAX) stock numbers are included where available following citation(s). Mouseline-originating articles are included following citation(s) if the citation is not the originating article. N.d. = no difference between healthy control and ALS.

Feature	Affected in ALS Patients?	Recapitulated in Mouse Models?
Reduced body weight	Yes [10,11,13,20,22]	Yes C9ORF72 Tg AAV (GGGGCC)66 [23] C9ORF72 BAC Tg [24] (JAX 029099) FUS WT Tg [25] (JAX 017916) FUS WT Tg [26] (JAX 027898) FUS ΔRRMcyt R522G Tg [27] FUS R521G Tg [26] (JAX 028021) MATR3 S85C KI [28] SOD1 T116X Tg [29] SOD1 G37R Tg [30] (JAX 008229) SOD1 G93A Tg [31] (Gurney et al., JAX 004435) SOD1 G86R Tg [32] (Ripps et al., JAX 005110) TDP-43 WT Tg [33] (JAX 016201) TDP-43 WT Tg [34] TDP-43 WT Tg [35] (JAX 016608) TDP-43 WT Tg [36] (JAX 031609) TDP-43 A315T Tg [33] (JAX 016143) TDP-43 A315T Tg [37] (JAX 010700) TDP-43 M337V Tg [33] TDP-43 M337V Tg [38] TDP-43 M337V Tg [39] (JAX 017604) TDP-43 N390D KI [40] No C9ORF72 BAC Tg [41] (JAX 030222) C9ORF72 BAC Tg [42] JAX (023088, 023099) TDP-43 A315T Tg [43] (Stallings et al.) (increased when compared to control) TDP-43 A315T KI [40] TDP-43 M337V KI [44] TDP-43 Q101X ENU [45] (JAX 019899) TDP-43 Q331K KI [46] (JAX 031345) TDP-43 Q331K Tg [47] (Arnold et al., JAX 017933) (increased when compared to control) See other mutant mice in this review paper [48]
Elevated resting energy expenditure	Yes [5,6,21]	Yes SOD1 G93A Tg [31,49] (Gurney et al., JAX 004435) SOD1 G86R Tg [32] (Ripps et al., JAX 005110) No SOD1 G93A Tg [50] (Gurney et al., JAX 002726)TDP-43 A315T Tg [43] (Stallings et al.) (n.d. when normalized to weight)
Reduced fat-free mass (or reduced muscle mass)	Yes [20,51,52]	Yes SOD1 G93A Tg [31] (Gurney et al., JAX 004435) TDP-43 A315T Tg [43] (Stallings et al.) TDP-43 Q331K Tg [47] (Arnold et al., JAX 017933) TDP-43 N390D KI [40] No TDP-43 A315T KI [40]
Reduced fat mass	Yes [20,51,52] N.d. [53]	Yes SOD1 G86R Tg [32] (Ripps et al., JAX 005110) SOD1 G93A Tg [31] (Gurney et al., JAX 004435) No TDP-43 A315T Tg [43] (Stallings et al.) (increased)
Dyslipidemia	Yes [54,55,56,57,58,59] N.d. [60] See more at [61,62]	Yes FUS WT Tg [63] (Mitchell et al., JAX 017916) SOD1 G93A Tg [64] (specific line or JAX not mentioned; B6SJL background) SOD1 G93A Tg [65] (Gurney et al., JAX 002726) SOD1 G86R Tg [66] (Ripps et al., JAX 005110) No TDP-43 A315T Tg [43] (Stallings et al.)
Reduced glucose utilization	Yes, in CNS [67]	Yes SOD1 G93A Tg [49,68] in muscles, CNS, respectively (Gurney et al., JAX 004435) SOD1 G86R Tg [69] in muscles (Ripps et al., JAX 005110) SOD1 G93A Tg [70,71] in CNS (Gurney et al., JAX 002726) TDP-43 A315T Tg [43] in muscles (Stallings et al.) TDP-43 A315T Tg [72] in CNS (Wegorzewska et al., JAX 010700) No SOD1 G86R Tg [32] (Ripps et al., JAX 005110) (increased glucose clearance in various tissues, including muscle and CNS)
Liver dysfunction	Yes [73,74]	SOD1 G93A Tg [75] (Gurney et al., JAX 002726)
Pancreas dysfunction	Yes [76]	SOD1 G93A Tg [77] (Gurney et al., JAX 004435)
Degeneration of hypothalamus	Yes [78,79,80] N.d. [81]	Yes SOD1 G93A Tg [82] (Gurney et al., JAX 002726) No FUS ΔNLS KI [82] (Scekic-Zahirovic et al.)
Food/caloric intake	Reduced [15,21,83]	Increased FUS ΔNLS KI [84] (Scekic-Zahirovic et al.) (after fasting) SOD1 G86R Tg [32,84] (Ripps et al., JAX 005110) (no fasting; after fasting, respectively) TDP-43 A315T Tg [84] (Wegorzewska et al., JAX 010700) (after fasting) TDP-43 Q331K KI [46] (JAX 031345) SOD1 G93A Tg [32] (Gurney et al., JAX but unclear which background) (n.d.) TDP-43 A315T Tg [43] (Stallings et al.) (n.d. when normalized to weight)

## Data Availability

Not applicable.

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
