# Peer review of "Evidence of Metabolic Dysfunction in Amyotrophic Lateral Sclerosis (ALS) Patients and Animal Models"

_biomolecules, 2023, doi:10.3390/biom13050863_

Round 1

Reviewer 1 Report

In this review, the authors focus on metabolic disturbances in ALS. Loss of body weight in ALS further strengthens this. However, in this context, lipidomics seems to be more relevant to the disease which authors hardly touched and missed important references such as DOI:10.1038/s41598-017-17389-9, 10.1038/s41598-019-48059-7,10.1038/s41598-021-92112-3. Regarding the ALS clinical features, can they make a table with references that will further confirm there Fig1.

Minor English corrections needed.

Author Response

Comment: In this review, the authors focus on metabolic disturbances in ALS. Loss of body weight in ALS further strengthens this. However, in this context, lipidomics seems to be more relevant to the disease which authors hardly touched and missed important references such as DOI:10.1038/s41598-017-17389-9, 10.1038/s41598-019-48059-7,10.1038/s41598-021-92112-3. Regarding the ALS clinical features, can they make a table with references that will further confirm there Fig1.

Our response: We appreciate your comments and suggestions as lipidomics studies provide important clues to dysregulated metabolism in ALS. As suggested, we added the suggested above three references (DOI:10.1038/s41598-017-17389-9, 10.1038/s41598-019-48059-7, 10.1038/s41598-021-92112-3) of lipidomics studies in Table 1 under “Dyslipidemia” and in the text of our manuscript (line 164-171). As there are more comprehensive, recent reviews on dyslipidemia and lipidomics studies in ALS, we have provided a brief introduction to this topic while guiding towards these reviews (references 61 and 62 in line 163 in the manuscript) for a more detailed looked at lipidomics in ALS.

Reviewer 2 Report

The manuscript by Maksimovic et al addresses a very interesting hot topic in the field of ALS research, both from the perspective of identifying molecular mechanisms involved in the pathology and also in terms of possible new promising therapeutic approaches for ALS patients.

Overall, the manuscript cites the most relevant literature supporting the inolvement of metabolic dysfunctions (both in the CNS and in peripheral tissues) in ALS animal models and in patients. Table 1 presents a pretty thoughrough overview of current evidence both at clinical and preclinical level, however I think that the organization of the manuscript might benefit of few rearrangements, listed below, to better highlight critical concepts in this field.

1. Table 1, second column, is not clear. The title is presented as a question ("Affected in ALS patients?"), however in the table the authors fail to clarify whether the reported references confirm/or not the presence of that specific dysfunction in ALS patients. This aspect should be fixed. Moreover, for each of the reported metabolic dysfunctions, the authors should clarify whether that dysfuction was reported early in the disease process (diagnosis or onset for patients; pre-symptomatic or onset for ALS animal models) or late in the disease (i.e. end stage). I believe that this is an important aspect that should be kept in mind also throughout the manuscript, in each dedicted paragraph when the authors describe the different metabolic dysfunctions. In fact, for instance, if some changes are observed at the end stage of the disease, this is most likely to be a consequence of the neurodegeneration, or even an effect associated with cachexya, whereas, on the other hand, if a deficit is reported early in the pathological process, this is likely to be an active contributor.

2. Paragraph on BMI. Connected with my previous comment, in this paragraph the authors highlight some literature evidence where researchers try to draw a correlation between BMI and disease severity and overall survival. However, the authors also discuss the possible involvement of dysphagia as contributor to altered BMI. Actually, dysphagia appears to be mainly a consequence of muscle wasting rather than a major contributor to the disease process. The authors should address this aspect.

3. Conflicting results. The authors mention some conflicting results, such as in lines 165-167, 187-188, 195-199. The authors should add comments (in each specific paragraph or in the conclusion) trying to identify an explanation for such incosistences or point to new evidence that could provide an hypothesis for reconciling such inconsistences.

Author Response

Comment #1: Table 1, second column, is not clear. The title is presented as a question ("Affected in ALS patients?"), however in the table the authors fail to clarify whether the reported references confirm/or not the presence of that specific dysfunction in ALS patients. This aspect should be fixed. Moreover, for each of the reported metabolic dysfunctions, the authors should clarify whether that dysfuction was reported early in the disease process (diagnosis or onset for patients; pre-symptomatic or onset for ALS animal models) or late in the disease (i.e. end stage). I believe that this is an important aspect that should be kept in mind also throughout the manuscript, in each dedicted paragraph when the authors describe the different metabolic dysfunctions. In fact, for instance, if some changes are observed at the end stage of the disease, this is most likely to be a consequence of the neurodegeneration, or even an effect associated with cachexya, whereas, on the other hand, if a deficit is reported early in the pathological process, this is likely to be an active contributor.

Our response #1: We thank Reviewer 2 for helpful suggestions and comments. As suggested, we have corrected the second column in Table 1 by adding Yes or n.d. (not determined) answer to the title “Affected in ALS patients?” (pages 2-4). We also thank for suggestions on discussing aspects of metabolic dysfunction being a cause or consequence of disease and have included a statement in the conclusion section referring to this in the manuscript in lines 397-405. In this statement, we have specified that the majority of papers examined patients at a mid-disease stage (~2 years disease duration) or mice at symptomatic stage and added comments on the importance of timing of study to illuminate our understanding of disease pathogenesis. Details of whether the study examined patients at diagnosis or pre-symptomatic stage in mice (where applicable) are already mentioned in the manuscript so no edits have been made in this regard.

Comment #2:  Paragraph on BMI. Connected with my previous comment, in this paragraph the authors highlight some literature evidence where researchers try to draw a correlation between BMI and disease severity and overall survival. However, the authors also discuss the possible involvement of dysphagia as contributor to altered BMI. Actually, dysphagia appears to be mainly a consequence of muscle wasting rather than a major contributor to the disease process. The authors should address this aspect.

Our response #2: As suggested, we have included more specifics on dysphagia being an early symptom in bulbar-onset ALS, and late for spinal-onset, and its relation to reduced caloric intake in our manuscript in lines 62-65.

Comment #3: Conflicting results. The authors mention some conflicting results, such as in lines 165-167, 187-188, 195-199. The authors should add comments (in each specific paragraph or in the conclusion) trying to identify an explanation for such incosistences or point to new evidence that could provide an hypothesis for reconciling such inconsistences.

Our response #3: As suggested, we have included proposed reasons to conflicting results where reasonable. In regards to conflicting results in dyslipidemia studies (corresponding to lines 165-167 in our old manuscript), we added our explanation in lines 179-185 in the revised manuscript. In regards to conflicting results in AMPK activation (corresponding to lines 187-188 in our old manuscript), we added that the inconsistencies are due to differences in mouse strain background and sexes used for experiments in examining liver AMPK activation in lines 205-207 in our revised manuscript. In regards to the conflicting results in pancreatic β-cells (corresponding to lines 195-199 in our old manuscript), our conclusion statement for that section reflects this issue in lines 219-221 in our revised manuscript.